# Colony-stimulating factor 3 signaling in colon and rectal cancers: Immune response and CMS classification in TCGA data

**Apryl S. Saunders**[1], **Dawn E. Bender**[1], **Anita L. Ray**[1], **Xiangyan Wu**[2,3], **Katherine T. Morris**[1]*

**1** Department of Surgery, The University of Oklahoma Health Sciences Center, Oklahoma City, Oklahoma, United States of America, **2** Academy of Integrative Medicine, Fujian University of Traditional Chinese Medicine, Fuzhou, Fujian, China, **3** Fujian Key Laboratory of Integrative Medicine in Geriatrics, Fujian University of Traditional Chinese Medicine, Fuzhou, Fujian, China

* katherine-morris@ouhsc.edu

**Data Availability Statement:** The datasets generated or analyzed during this study are openly available in The Cancer Genome Atlas at https://www.cancer.gov/tcga.

## Abstract

Colorectal cancer is the 2nd leading cause of cancer-related deaths in the world. The mechanisms underlying CRC development, progression, and resistance to treatment are complex and not fully understood. The immune response in the tumor microenvironment has been shown to play a significant role in many cancers, including colorectal cancer. Colony-stimulating factor 3 (CSF3) has been associated with changes to the immune environment in colorectal cancer animal models. We hypothesized that CSF3 signaling would correlate with pro-tumor tumor microenvironment changes associated with immune infiltrate and response. We utilized publicly available datasets to guide future mechanistic studies of the role CSF3 and its receptor (CSF3R) play in colorectal cancer development and progression. Here, we use bioinformatics data and mRNA from patients with colon (n = 242) or rectal (n = 92) cancers, obtained from The Cancer Genome Atlas Firehose Legacy dataset. We examined correlations of CSF3 and CSF3R expression with patient demographics, tumor stage and consensus molecular subtype classification. Gene expression correlations, cell type enrichment, Estimation of STromal and Immune cells in MAlignant Tumor tissues using Expression data scores and Gene Ontology were used to analyze expression of receptor and ligand, tumor microenvironment infiltration of immune cells, and alterations in biological pathways. We found that CSF3 and CSF3R expression is highest in consensus molecular subtype 1 and consensus molecular subtype 4. Ligand and receptor expression are also correlated with changes in T cell and macrophage signatures. CSF3R significantly correlates with a large number of genes that are associated with poor colorectal cancer prognosis.

## Introduction

Colorectal cancer (CRC) is the second most common cause of cancer-related deaths and the third most commonly diagnosed cancer world-wide [1]. CRC risk, progression, and outcome

**Funding:** This work was funded by the American Cancer Society under Grant MRSG-15-136-01-CCE (KTM) https://urldefense.proofpoint.com/v2/url?u=http-3A__www.cancer.org&d=DwIGaQ&c=VjzId-SM5S6aVB_cCGQ0d3uo9UfKByQ3sI6Audoy6dY&r=47E30vBjgUhCUgf1o9ZUiNABBuJGlF2-pQOCqm470bM&m=

are affected by sex, age, and environmental factors, and CRCs are molecularly and biologically diverse [2]. CRCs are classified into four distinct consensus molecular subtypes (CMS) based on distinct molecular, clinical, and mRNA expression features and are labeled as CMS1-CMS4 [3]. These subtypes are associated with different tumor microenvironment (TME) makeup and outcome: CMS1 (microsatellite instable (MSI) immune), CMS2 (canonical), CMS3 (metabolic), and CMS4 (mesenchymal). Due to the substantial mortality rate of CRC and molecular diversity of these different subtypes, there is a critical need for targeted therapeutics for improving patient care.

In the local TME, interactions between malignant and non-malignant cells, such as CD8 cytotoxic T cells and other immune infiltrates, play significant roles throughout the course of CRC, from initial development to final outcome [4]. Current treatments for CRC include surgery, chemotherapy, radiation, and targeted biologic agents. Although anti-tumor immune responses are associated with improved survival, current immunotherapies are not as successful in most CRC as in other cancers such as melanoma or lung cancer, although there are effective responses in microsatellite instable tumors [5–8].

Colony-stimulating factor 3 (CSF3) is a cytokine used clinically for promoting the production and release of bone marrow-derived hematopoietic stem cells and granulocytes [9]. However, CSF3 has been shown to affect additional aspects of the innate and adaptive immune systems [10–14]. CSF3 and CSF3 receptor (CSF3R) are highly expressed in CRC and other tumors, compared to normal tissues from the same organ [10]. CSF3 is an important regulator of CRC, increasing pro-tumor behavior in tumor and immune cells. CSF3 increases tumor cell proliferation, migration, and the proportion of stem-like cells in culture in CRC [10]. Furthermore, inhibition of CSF3 with functional antibody resulted in significant shifts in both the innate and adaptive immune compartments. CSF3 inhibition increased T cell infiltration, and decreased neoplasm development by 75% in the AOM/DSS model of CRC development [14]. These findings suggest potential pro-tumor effects of CSF3 signaling that are of particular concern because recombinant CSF3 is currently used to prevent and treat febrile neutropenia secondary to chemotherapy. The effects of this treatment on tumor and immune responses in CRC patients are unknown, but the potential risk of creating a pro-tumor TME increases the importance of understanding the role of this pleiotropic cytokine in cancer.

Currently, it is unknown whether CSF3 and CSF3R expression are associated with patient demographics, stage at presentation, location of tumor, mutational status, CMS subtype, or distinct patterns of immune infiltrates. Therefore, we examined publicly available datasets using bioinformatics data analysis techniques to answer those questions. The Cancer Genome Atlas (TCGA) data analysis revealed that CSF3 and CSF3R are associated with a large number of changes in the tumor microenvironment of CRC. Additionally, CSF3R expression is upregulated in CMS subtypes 1 and 4 in colon cancer (CC) and rectal cancer (RC). Higher expression of CSF3R in both CC and RC was found to correlate strongly with higher scores using the "estimation of stromal and immune cells in malignant tumor tissues using expression data" (ESTIMATE) analysis. Cell-type enrichment analysis revealed that CSF3 negatively correlates with naïve CD8 signatures, while CSF3R positively correlates with both T regulatory (Treg) and macrophage infiltration. Gene ontology (GO) biological process enrichment analysis showed many similarities between CC and RC in the CSF3 pathway. Relevant pathways include T cell regulation, macrophage activation, and ERK/MAPK signaling. We found correlations of CSF3 and CSF3R with multiple gene signatures associated with poor outcome. These analyses provide critical insight into diverse molecular pathways and immune response in the TME of CRC, and suggest CSF3 as a possible therapeutic target.

## Materials and methods

### The Cancer Genome Atlas (TCGA) colon and rectal cancer dataset

Publicly available datasets of CC and RC (n = 92) patients were obtained from TCGA using either cBioPortal for Cancer Genomics or Xena browser (https://xenabrowser.net) [15, 16]. These datasets included de-identified patient demographics and RNA Seq data.

### Clinicopathologic characteristics

Demographic data from CC (n = 242) and RC (n = 92) patients were obtained from TCGA using cBioPortal for Cancer Genomics [15, 17]. Because all but the earliest stage rectal cancers are most commonly treated with neoadjuvant chemoradiation therapy and preoperative chemotherapy has been shown to affect the molecular classification of colorectal cancers, we analyzed the two separately [18]. Potential demographic associations were assessed using high and low CSF3 and CSF3R expression levels stratified by the mean (TCGA Firehose Legacy RNA Seq V2 RSEM). Significance was determined by Fisher's exact test using IBM SPSS Statistics. A p-value of <0.05 was considered statistically significant and the strength of the correlation was described as follows: 0.00–0.19 "very weak", 0.20–0.39 "weak", 0.40–0.59 "moderate", 0.60–0.79 "strong", 0.80–1.0 "very strong" [19, 20].

### Tumor classification correlations

RNA-seq data of CC (n = 329) and RC (n = 94) tumors and related clinical information was acquired from TCGA using Xena browser [16]. CMS subtypes were determined using the CMSCaller R packages [21].

### ESTIMATE

ESTIMATE algorithms were used to calculate ESTIMATE, stromal, and immune scores from TCGA CC and RC tumor datasets using RNA Seq V2 platform [22]. These were then compared with CSF3 and CSF3R mRNA expression data (Genentech, Nature 2012 [23]) retrieved from cBioPortal for Cancer Genomics [15]. Correlation analysis was performed using GraphPad Prism 8.

### Immune cell signatures

Immune cell correlation analysis of CSF3 and CSF3R was performed on TCGA data using the immunedeconv [24] package in R and xCell method analysis via TIMER2.0 (http://timer.cistrome.org/) [25–27]. Samples were adjusted for purity where recommended and only correlations with Spearman's p-value < 0.05 are reported.

### Gene Ontology (GO)

Biological process enrichment analysis of genes correlated to CSF3 and CSF3R expression was evaluated with Spearman's correlation test. Reported correlations with CSF3 are those with $\rho$ correlation coefficients > 0.3. Due to the high number of positively correlated genes with CSF3R expression, we used only those with a strong correlation coefficient of > 0.7 to calculate pathway enrichment. Statistical analysis performed using Panther GO-slim Overrepresentation Test (Released 20200925), test type: Fisher [28]. Hierarchal categories of pathways were removed to highlight most specific processes. All p-values are < 0.001.

## Gene expression correlations

mRNA expression correlation analysis of CSF3 and CSF3R was performed on TCGA CC (n = 242) and RC (n = 92) (Firehose Legacy RNA Seq V2 RSEM) datasets using cBioPortal for Cancer Genomics [15]. Scatterplots and correlations were taken directly from cBioPortal.

# Results

## Neither CSF3 nor CSF3R are correlated with standard clinicopathologic characteristics

Table 1 shows the clinicopathologic data for patients and tumors, stratified by CSF3 and CSF3R expression. For this part of the study, we used the TCGA Legacy Firehose dataset, consisting of RNA-seq data from CC (n = 242) or RC (n = 92) tumors. Tumors were stratified into high or low expression groups by mean expression levels of CSF3 or CSF3R. Demographic information stratified by CSF3 and CSF3R expression is shown in Table 1. The 334 patients consisted of 186 males (56%) and 148 females (44%), aged 31–90 years (median 66). We hypothesized that CSF3 and CSF3R would correlate with male sex and increasing age. The data did not support this hypothesis, with minimal age differences between patients with high and low CSF3 or CSF3R expressing tumors. While not meeting criteria for statistical significance, a higher percentage of CC patients with high CSF3 were male (67% vs 33%, p-value = 0.112). It is possible there is a small biological difference in CSF3 expression in the

**Table 1. Clinical and demographic characteristics of CC and RC patients.**

| | CC | | | | | | RC | | | | | |
|---|---|---|---|---|---|---|---|---|---|---|---|---|
| | Low CSF3 | High CSF3 | p- value | Low CSF3R | High CSF3R | p- value | Low CSF3 | High CSF3 | p- value | Low CSF3R | High CSF3R | p- value |
| **Variables** | | | | | | | | | | | | |
| **Age: mean (SD)** | 65(13) | 66(13) | 0.389 | 64(14) | 67(12) | 0.110 | 63(12) | 64(13) | 0.621 | 64(12) | 62(13) | 0.559 |
| | n = 191 | n = 51 | | n = 160 | n = 82 | | n = 66 | n = 26 | | n = 58 | n = 34 | |
| **Sex: n (%)** | | | 0.112 | | | >0.999 | | | 0.487 | | | 0.137 |
| Female | 89(47) | 17(33) | | 70(44) | 36(44) | | 32(48) | 10(38) | | 30(52) | 12(35) | |
| Male | 102(53) | 34(67) | | 90(56) | 46(56) | | 34(52) | 16(62) | | 28(48) | 22(65) | |
| **Stage: n (%)** | | | 0.514 | | | 0.209 | | | >0.999 | | | 0.176 |
| I-II | 107(58) | 25(52) | | 83(54) | 49(63) | | 27(46) | 11(46) | | 27(52) | 11(35) | |
| III-IV | 77(42) | 23(48) | | 71(46) | 29(37) | | 32(54) | 13(54) | | 25(48) | 20(65) | |
| **Tumor: n (%)** | | | 0.222 | | | 0.727 | | | 0.556 | | | 0.582 |
| T1-T2 | 38(20) | 6(12) | | 28(18) | 16(20) | | 11(17) | 6(23) | | 12(21) | 5(15) | |
| T3-T4 | 152(80) | 45(88) | | 131(82) | 66(80) | | 54(83) | 20(77) | | 45(79) | 29(85) | |
| **Node: n (%)** | | | 0.524 | | | 0.338 | | | 0.817 | | | 0.271 |
| N0 | 115(60) | 28(55) | | 91(57) | 52(63) | | 29(46) | 11(42) | | 28(50) | 12(36) | |
| N1-N2 | 76(40) | 23(45) | | 69(43) | 30(37) | | 34(54) | 15(58) | | 28(50) | 21(64) | |
| **Metastasis: n (%)** | | | 0.183 | | | 0.693 | | | 0.736 | | | 0.341 |
| M0 | 131(85) | 34(76) | | 109(81) | 56(85) | | 46(85) | 18(82) | | 42(88) | 22(79) | |
| M1 | 24(15) | 11(24) | | 25(19) | 10(15) | | 8(15) | 4(18) | | 6(12) | 6(21) | |
| **Tumor Site: n (%)** | | | 0.405 | | | 0.477 | | | - | | | - |
| Left | 75(42) | 16(34) | | 58(38) | 33(43) | | - | - | | - | - | |
| Right | 105(58) | 31(66) | | 93(62) | 43(57) | | - | - | | - | - | |

TCGA data was used to profile patient and tumor characteristics for CC (n = 242) and RC (n = 92). Analysis of age was performed using Student's t test for significance; all other analyses utilized Fisher's exact test.

tumors from men and women that is not seen in this sample size, but the relevance of that is unclear. Higher tumor or nodal stages did not demonstrate significant enrichment of CSF3 or CSF3R expression, which was surprising given that smaller studies have found higher expression of both ligand and receptor in higher T and N stage patients [10]. This demonstrates the importance of using larger publicly available datasets to confirm or refute findings from smaller studies. While not significant, a higher percentage of CC patients who had metastatic disease at time of diagnosis had primary tumors in which CSF3 was expressed above the mean (24% high CSF3 as compared to 15% low CSF3, p-value = 0.183). While this finding does not meet the preset criteria for statistical significance, it will be important to examine larger datasets as they become available to confirm a Type II error has not been made. We also evaluated the relationship between receptor and ligand expression in these tumors. We found a weak to moderate correlation between CSF3 and CSF3R expression (CC Spearman ρ = 0.227, p-value = 3.774e-4; RC Spearman ρ = 0.417, p-value = 3.505e-5) (S1 Fig).

## Elevated CSF3R is associated with CMS1 and CMS4 subtypes

Mean CSF3 and CSF3R gene expressions were compared between the samples in the 4 CRC subtypes: CMS1 (n = 49), CMS2 (n = 76), CMS3 (n = 43) and CMS4 (n = 97) in CC; and CMS1 (n = 10), CMS2 (n = 20), CMS3 (n = 18) and CMS4 (n = 31) in RC (Fig 1A). CSF3 levels

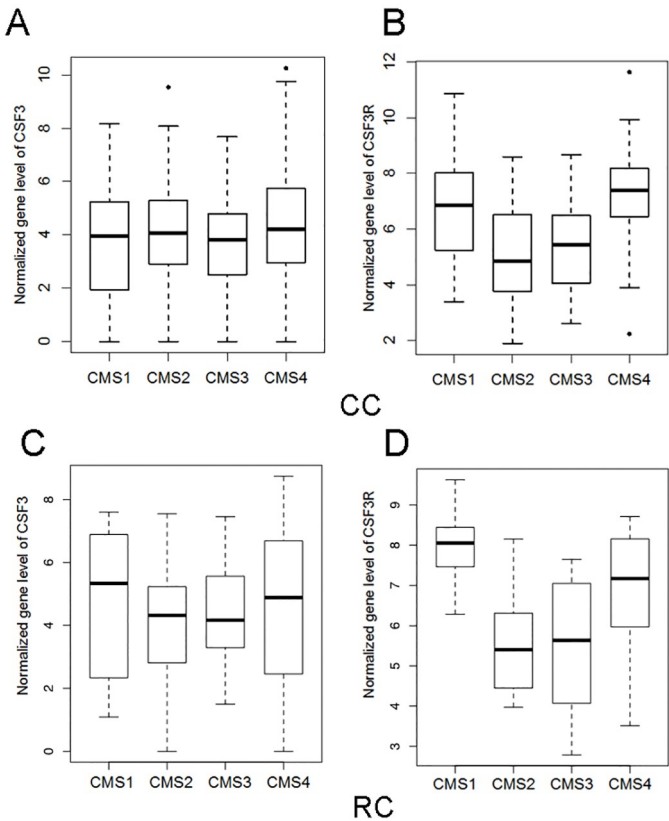

**Fig 1. CSF3 signaling is most upregulated in CMS1 and CMS4 in CRC.** CMS categories were determined on CC and RC tumors from the TCGA dataset using the CMScaller package in R. Box and whisker plots displaying (A) CSF3 expression and (B) CSF3R expression levels in CC and (C, D) RC tumors normalized to non-malignant tissue in each of the CMS subcategories.

were similar in the four CRC subtypes in CC. However, CSF3R expression was higher in both the CMS1 and CMS4 classifications (Fig 1B). In RC, both CSF3 and CSF3R were higher in expression in both CMS1 and CMS4 (Fig 1C and 1D). This analysis suggests that CSF3 signaling may play a greater role in the development, progression and treatment response in CMS1 and CMS4 CRC tumors.

## Higher CSF3R expression is correlated with an adverse immune infiltrate and stromal environment

Stromal and immune infiltration in the TME plays an important role in the progression of solid tumors. Higher ESTIMATE scores have been correlated with poor prognosis in multiple cancers, with higher scores associated with shorter overall survival in CRC patients [29, 30]. Utilizing this algorithm with the TCGA CC and RC data, we calculated correlation coefficients between ESTIMATE scores and CSF3 ligand and receptor expression (all reported data has p-values < 0.0001). CSF3R strongly correlates with stromal (Spearman $\rho$ = 0.6182), immune (Spearman $\rho$ = 0.7201), and ESTIMATE scores in CC (Fig 2A–2C). In RC, CSF3R expression has a moderate correlation with the stromal score (Spearman $\rho$ = 0.5994), and strongly correlates with immune (Spearman $\rho$ = 0.6602) and ESTIMATE scores (Spearman $\rho$ = 0.7084) (Fig 2D–2F). These correlations suggest that higher CSF3R expression is a characteristic of multiple gene signatures associated with pro-tumor immune environments in CRC patients.

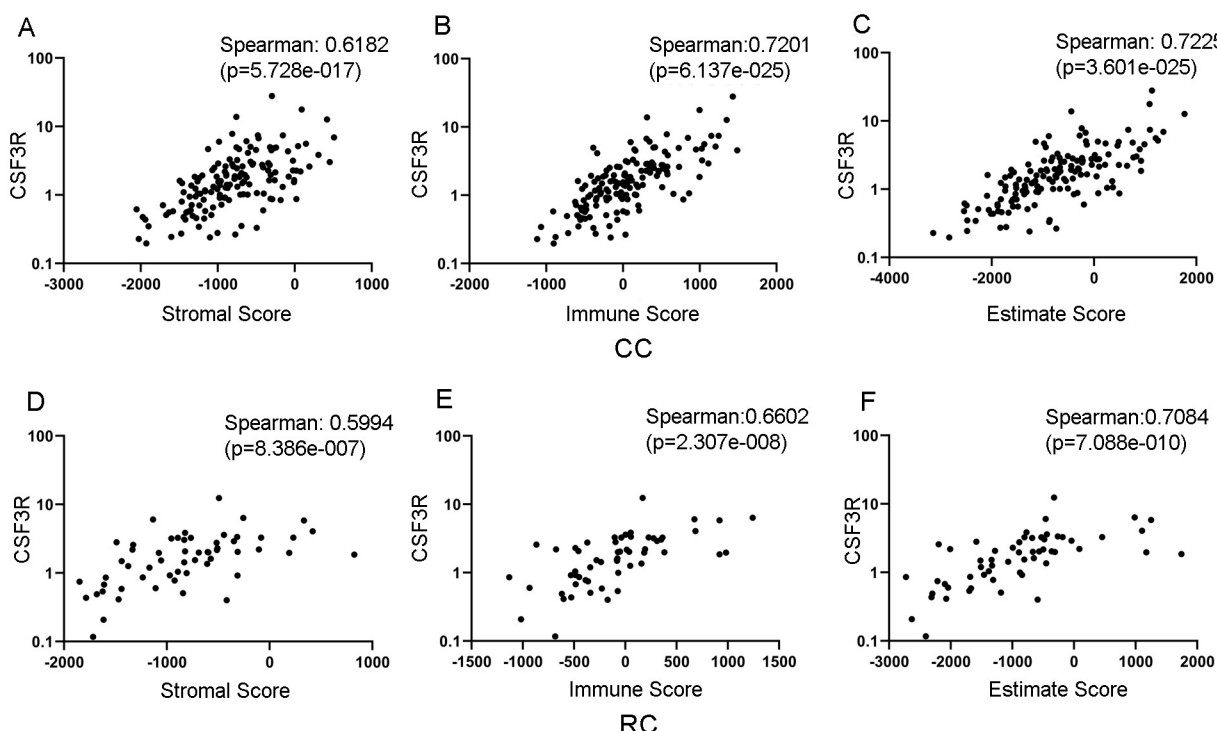

**Fig 2. CSF3 ligand and receptor expression correlate with immune and stromal infiltration in the TME.** Scatterplots depicting CSF3R mRNA expression demonstrate strong correlations with (A) Stromal, (B) Immune and (C) ESTIMATE scores in CC. CSF3 expression moderately correlates with (D) Stromal score, and strongly correlates with (E) Immune and (F) ESTIMATE Scores. mRNA microarray data from TCGA Nature 2012 dataset was compared to ESTIMATE scores taken from https://bioinformatics.mdanderson.org/estimate with matched patient samples.

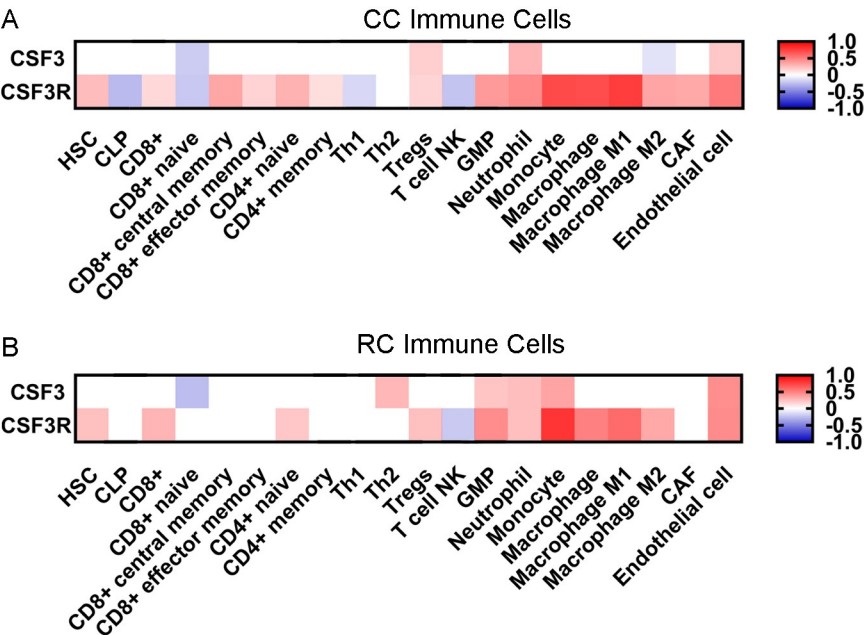

**Fig 3. CSF3 signaling correlates with distinct changes in immune cell composition of the TME.** Heatmaps representing xCell gene set enrichment analysis (GSEA) of correlations between CSF3 or CSF3R expression and infiltration of specific immune cell subtypes in (A) CC and (B) RC tumors. Red (positive) and blue (negative) scale represents Spearman's correlation with evaluated with purity adjustment, p-values > 0.05 considered as no correlation. White squares denote correlations for which p-values were not significant. Data acquired from timer.cistrome.org.

## Elevated CSF3 and CSF3R expression correlate with distinct immune cell type infiltrates

CSF3 signaling is associated with changes in the immune infiltrate within the TME and demonstrates strong correlation with gene signatures associated with pro-tumor immune responses [14]. Therefore, we analyzed TCGA CC and RC data for immune cell signatures. Cell type enrichment analysis of CSF3 and CSF3R expression uncovered strong correlations of CSF3R with multiple immune cell types in both CC and RC (Fig 3).

**T cells.** In CC, we see shifts in multiple T cell phenotypes, both towards pro-tumor and anti-tumor responses that correlate with CSF3 and CSF3R expression (Fig 3A). There is an increase of CD8 central memory cells (Spearman $\rho = 0.342$) and CD4 naïve cells (Spearman $\rho = 0.298$) associated with CSF3R expression. There is very weak but significant correlation between CSF3R and CD8 effector memory cells (Spearman $\rho = 0.171$), CD4 memory cells (Spearman $\rho = 0.128$), and Tregs (Spearman $\rho = 0.167$). Alternatively, there is a decrease in natural killer (NK) T cells (Spearman $\rho = -0.232$) and CD8 naïve cells (Spearman $\rho = 0.219$). In RC, the CSF3R associated enrichment of CD4 naïve cells (Spearman $\rho = 0.217$) and Tregs (Spearman $\rho = 0.243$) is observed, as well as the decrease of natural killer T cells (Spearman $\rho = -0.212$) (Fig 3B). Interestingly, we see fewer changes in T cell infiltration associated with CSF3 expression, limited to a decrease in CD8 naïve cells in CC and RC (Spearman $\rho = -0.201$ and 0.260, respectively), CC specific increase in Tregs (Spearman $\rho = 0.190$), and RC specific increase in Th2 (Spearman $\rho = 0.280$).

**Myeloid cells.** In addition to changes in T cell compartments, similar CSF3R associated increases in monocyte and macrophage signatures are observed in CC and RC. This includes monocytes (CC Spearman $\rho = 0.710$, RC Spearman $\rho = 0.793$) and both M1 (CC Spearman

ρ = 0.759, RC Spearman ρ = 0.578) and M2 (CC Spearman ρ = 0.352, RC Spearman ρ = 0.336) macrophages, with a stronger association for monocytes and M1 macrophages. Interestingly, CSF3 expression in CC is associated with a decrease in M2 macrophages. In addition, there is an increase in neutrophils in CC and RC with both CSF3 (CC Spearman ρ = 0.298, RC Spearman ρ = 0.252) and CSF3R (CC Spearman ρ = 0.455, RC Spearman = ρ 0.247). These data suggest that both CSF3 and CSF3R expression are associated with significant changes in the immune cell milieu within the TME of CRC.

## CSF3 and CSF3R correlate with T cell, macrophage, and ERK/MAPK signaling signatures

High CSF3 expression correlates with expression changes in a substantial number of genes. CSF3 expression correlated with 104 genes in CC and 495 genes in RC with a Spearman ρ > 0.3, and all with p-values <0.05. CSF3R had much stronger correlations with gene expression changes than with its ligand, therefore we used a more stringent cut-off. CSF3R expression in CC correlated with 277 genes and in RC correlated with 270 genes Spearman ρ > 0.7 and a p-value of < 0.05. These genes are listed in full in S1 Dataset.

Increased CSF3 and CSF3R gene expression correlates with changes in multiple pathways implicated in CRC (Figs 4 and 5). To determine additional affected pathways associated with increased CSF3 and CSF3R expression, we utilized gene enrichment analysis via the Gene Ontology Consortium (GO) (http://geneontology.org/). Of particular interest are signatures regulating T cell activation and proliferation, macrophage response, and immune chemotaxis.

A prodigious nearly 80-fold enrichment of genes in the macrophage activation pathway was identified with increased CSF3 expression. In addition, there was a more than 40-fold enrichment in genes involved in granulocyte chemotaxis and 17-fold increase in lymphocyte migration in CC (Fig 4A). Interestingly, CSF3R expression is correlated with a more than 20-fold enrichment in genes associated with positive and negative regulation of T cell proliferation in CC (Fig 4B). Lesser but still significant enrichment is seen in inflammatory response, innate immune response and cell chemotaxis as well.

CSF3 expression is associated with both positive and negative regulation of T cell activation and granulocyte chemotaxis and notably, angiogenesis in RC (Fig 5A). Similar to CC, genes that correlated with CSF3R expression revealed more than 15-fold enrichment in genes associated with positive and negative regulation of T cell proliferation and enrichment in genes associated with inflammatory response, innate immune response, and granulocyte chemotaxis (Fig 5B).

In addition, tumor and immune-associated mitogen activating protein kinase (MAPK) and extracellular signaling-regulated kinase (ERK) signaling pathways were enriched. These signaling pathways play complex roles in cellular proliferation, differentiation and apoptosis. We observed enrichment of genes correlated with CSF3 expression with positive regulation of ERK1 and ERK2 cascade in both CC (Fig 4A) and RC (Fig 5A). CSF3R expression was also associated with this pathway in RC tumors (Fig 5B), whereas positive regulation of MAPK pathway was identified in CC (Fig 4B). Identification of these often-intertwined pathways suggests a relationship between CSF3 signaling and cell proliferation and tumor progression.

## CSF3 and CSF3R expression are associated with regulators of invasion

There were invasion-associated chemokines correlated with CSF3 expression. CSF3 expression significantly correlates with CXCL5 (CC Spearman ρ = 0.6694, RC Spearman ρ = 0.7140), CXCL6 (CC Spearman ρ = 0.6030, RC Spearman ρ = 0.7022), and CXCL8 (CC Spearman ρ = 0.5546, RC Spearman ρ = 0.6748) in both CC and RC (Table 2). CSF3R has similar correlations

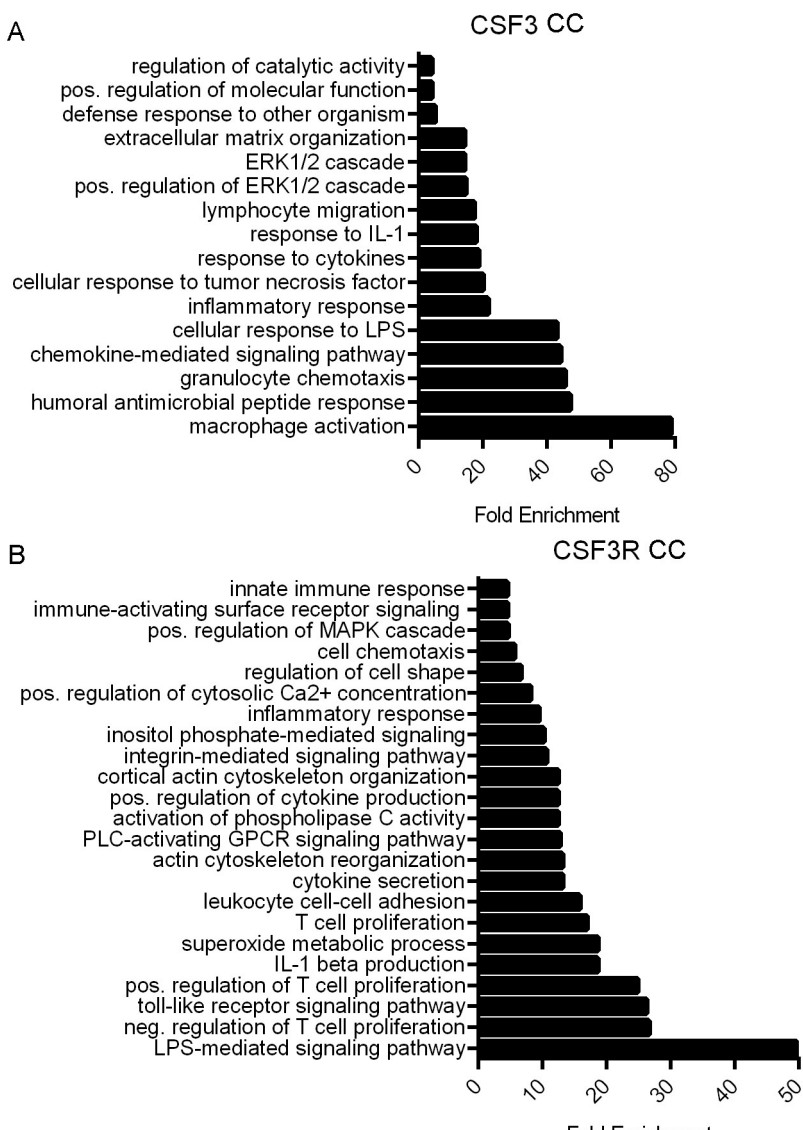

**Fig 4. CSF3 and CSF3R expression are associated with immune activity and cell signaling enrichment in CC.** Bar graphs indicating fold enrichment of pathways using Gene Ontology Consortium (GO) Panther GO-slim biological process analysis of genes correlated with increased (A) CSF3 expression with a Spearman's $\rho > 0.03$ and (B) CSF3R expression with a Spearman's $\rho > 0.07$. TCGA Firehose Legacy dataset analyzed via Fisher's exact test type with False Discovery Rate (FDR) correction.

with these chemokines. CXCL8 has the strongest correlation in CC (Spearman $\rho = 0.5798$), with CXCL5 (Spearman $\rho = 0.4353$) and CXCL6 (Spearman $\rho = 0.4669$) at slightly lower levels. RC shows a similar pattern (CXCL8 $\rho = 0.6685$, CXCL5 $\rho = 0.5891$, CXCL6 $\rho = 0.5038$). Scatterplots for these data are in Fig 6 (due to space considerations, additional scatterplots for other genes in Table 2 are in S2–S4 Figs). All correlations described have p-values < 0.01.

Because these cytokines have been associated with pro-tumor effects, we also looked for cytokines classically associated with CRC initiation and progression. CSF3 was found to have a significant correlation with IL-6 and IL-1. IL-6 is correlated with both CSF3 (CC Spearman $\rho = 0.69$, p = 7.66e-36; RC Spearman $\rho = 0.75$, p = 1.58e-17) and CSF3R (CC Spearman $\rho = 0.55$, p = 1.14e-20; RC Spearman $\rho = 0.71$, p = 1.71e-15). In CC and RC, CSF3 is positively

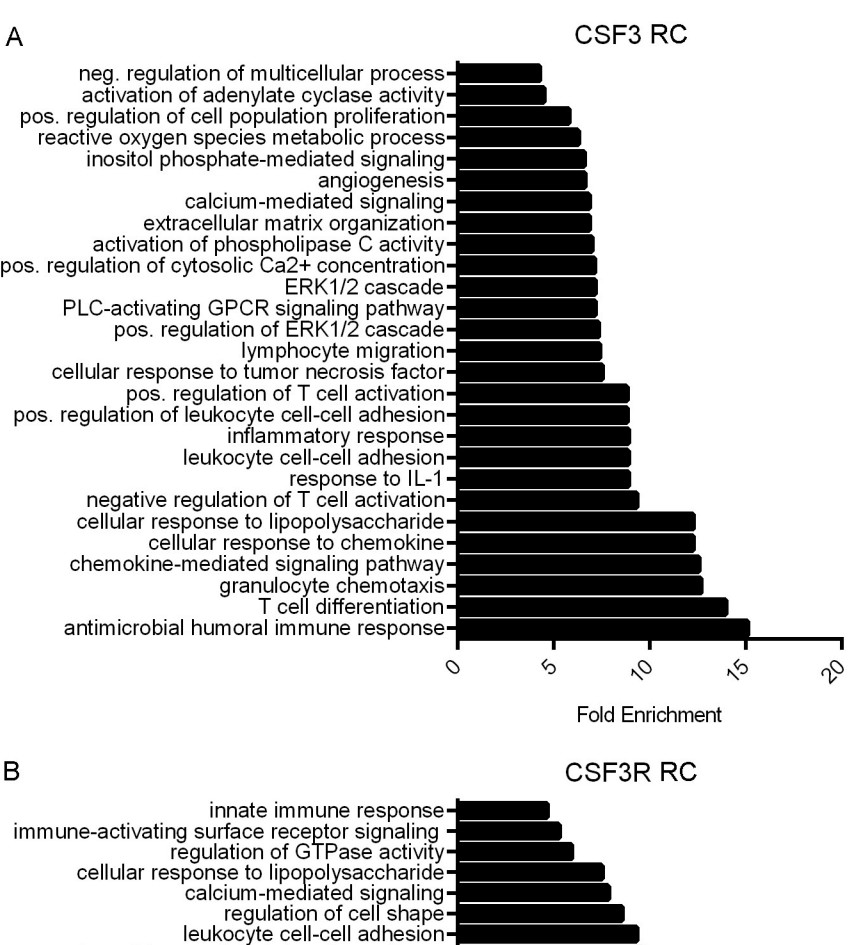

**Fig 5. CSF3 and CSF3R expression are associated with immune activity and cell signaling enrichment in RC.** Bar graphs indicating fold enrichment of pathways using Gene Ontology Consortium (GO) Panther GO-slim biological process analysis of genes correlated with increased (A) CSF3 expression with a Spearman's ρ > 0.03 and (B) CSF3R expression with a Spearman's ρ > 0.07. TCGA Firehose Legacy dataset analyzed via Fisher's exact test type with FDR correction.

**Table 2. CSF3 and CSF3R are associated with invasion-related gene expression.**

| | CC | | RC | |
|---|---|---|---|---|
| | CSF3 | CSF3R | CSF3 | CSF3R |
| **CXCL5** | 0.6694 | 0.4353 | 0.7140 | 0.5891 |
| **CXCL6** | 0.6030 | 0.4669 | 0.7022 | 0.5038 |
| **CXCL8** | 0.5546 | 0.5798 | 0.6748 | 0.6685 |
| **IL6** | 0.6921 | 0.5516 | 0.7454 | 0.7122 |
| **IL1a** | 0.4411 | 0.1699 | 0.4916 | 0.2805 |
| **IL1b** | 0.6064 | 0.5216 | 0.6937 | 0.6682 |
| **FPR1** | 0.2735 | 0.9226 | 0.4759 | 0.9318 |
| **FPR2** | 0.3611 | 0.8872 | 0.5883 | 0.8760 |
| **MMP1** | 0.5600 | 0.4986 | 0.7124 | 0.5299 |
| **MMP3** | 0.6965 | 0.3956 | 0.7727 | 0.4598 |
| **MMP10** | 0.5881 | 0.2462 | 0.6408 | 0.2756 |

CSF3 and CSF3R were analyzed for gene expression correlation with other genes from CC or RC tumors using cBioPortal for Cancer Genomics. Patterns of correlation with invasion-related chemokines, cytokines, formyl peptide receptors, and metalloproteinases are summarized here. All p-values < 0.01.

correlated with IL-1α and IL-1β (CC IL-1α Spearman ρ = 0.44, p = 6.08e-13, and IL-1β Spearman ρ = 0.61, p = 1.09e-25; RC IL-1α Spearman ρ = 0.49, p = 6.47e-7, and IL-1β Spearman ρ = 0.69, p = 1.78e-14). CSF3R has a correlation with IL-1β of ρ = 0.52 (p = 2.77e-18) and ρ = 0.67 (p = 3.40e-13) in CC and RC, respectively. It has a weak but significant correlation with IL-1α (Spearman ρ = 0.17, p = 8.081e-3) in CC. The correlation between CSF3R and IL-1α is slightly stronger in RC (Spearman ρ = 0.28, p = 6.769e-3).

Interestingly, both cancers showed significant correlations of formyl peptide receptors FPR1 and FPR2 with CSF3 and CSF3R. CSF3 had relatively weak correlations with FPR1 (CC Spearman ρ = 0.2735, RC Spearman ρ = 0.4759) and slightly stronger correlations with FPR2 (CC Spearman ρ = 0.3611, RC Spearman ρ = 0.5883). However, CSF3R had some of its strongest correlations with FPR1 (CC Spearman ρ = 0.9226, RC Spearman ρ = 0.9318) and FPR2 (CC Spearman ρ = 0.8872, RC Spearman ρ = 0.8760). Scatterplots for these correlations are available in S3 Fig.

In reviewing the correlations with the highest ρ values, we identified three MMP family proteins that were moderately to strongly correlated with CSF3 and CSF3R expression. We found that CSF3 expression significantly correlates with MMP1 (CC Spearman ρ = 0.5600, RC Spearman ρ = 0.7124), MMP3 (CC Spearman ρ = 0.6965, RC Spearman ρ = 0.7727), and MMP10 (CC Spearman ρ = 0.5881, RC Spearman ρ = 0.6408). These are particularly high correlations for CSF3. CSF3R also has significant weak to moderate correlations with these proteins. MMP1 has the highest ρ value (0.4986 in CC and 0.5299 in RC), followed by MMP3 (0.3956 CC, 0.4598 RC). MMP10 has a weak association with CSF3R expression (0.2462 in CC and 0.2756 in RC). MMP1, MMP3, and MMP1 can break down extracellular matrices and potentially contribute to tumor invasion [31–33].

Together, these protein families suggest that CSF3 and CSF3R expression may be characteristic of a number of pathways linked to invasion in CRC and other tissue types.

## Discussion

CSF3 is a pleiotropic cytokine that has significant effects on tumor and immune cells. We hypothesized that CSF3 and CSF3R would be associated with pro-tumor immune signatures

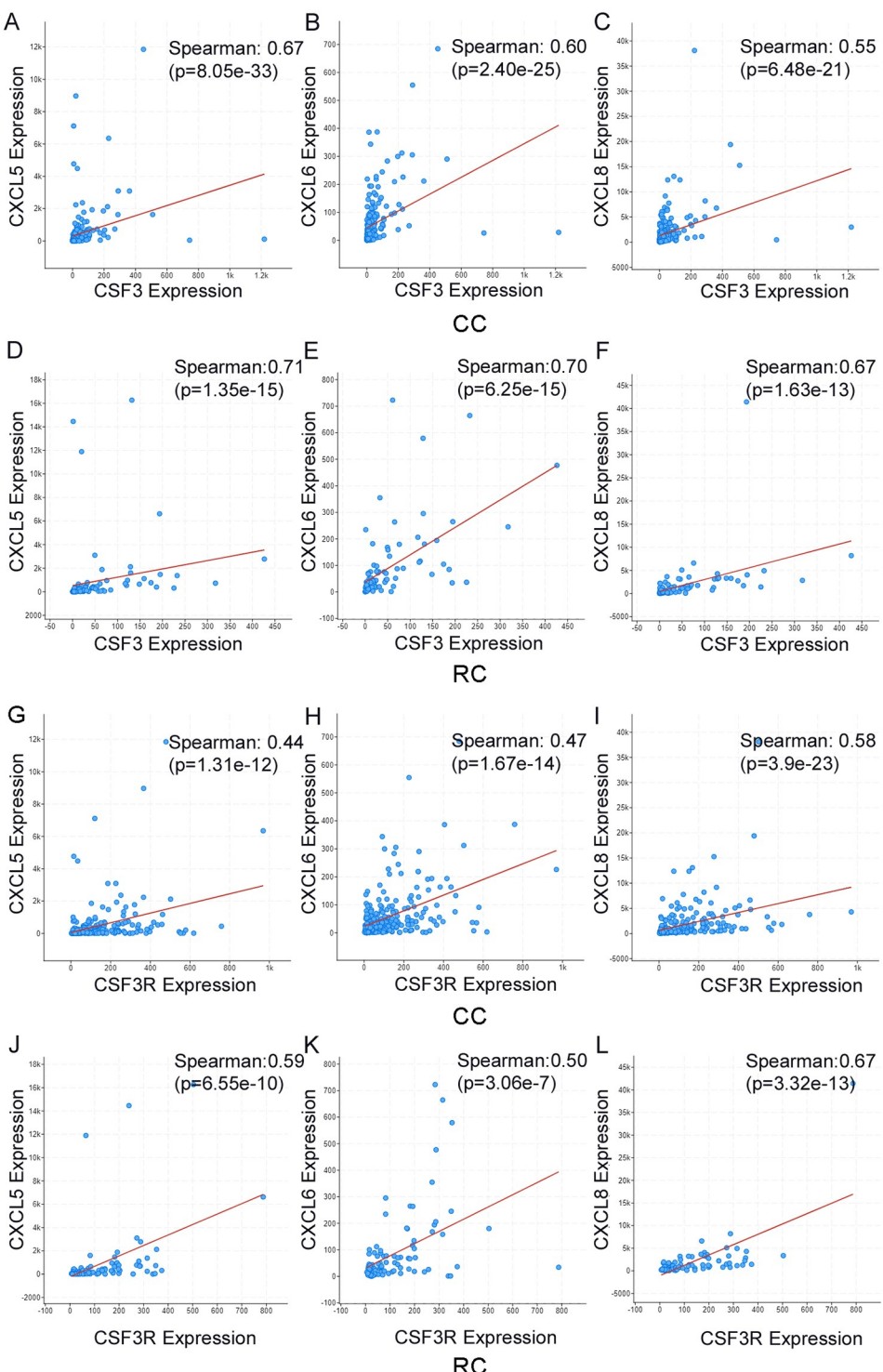

**Fig 6. CSF3 ligand and receptor expression is associated with several immune regulatory cytokines.** Scatterplots depicting Spearman's correlation between CXCL5, CXCL6, or CXCL8 and CSF3 in CC (**A-C**) and RC (**D-F**); and CXCL5, CXCL6, or CXCL8 and CSF3R in CC (**G-I**) and RC (**J-L**). Red line indicates linear regression.

and pathways. Using publicly available data and validated gene signatures, alongside an examination of significantly enriched genes, we looked at a broad overview of tumor phenotypes. Here, we show that expression of CSF3 and CSF3R are associated with CMS1 and CMS4 classifications, pro-tumor gene scores, and regulation of both innate and adaptive cells in the TME. These findings suggest potential associations of CSF3 signaling with how the TME is shaped and maintained, potentially affecting responses to immunotherapies [34]. CSF3 signaling is also associated with many of the tumor infiltrating immune populations, including T cells and macrophages, which are important regulators of the TME [35].

We found an association between CMS classification and CSF3R in both CC and RC, and a weaker association with the CSF3 ligand specific to RC. CMS1 and CMS4 have different immune profiles and prognoses [36]. CMS1, with an increase in cytotoxic lymphocyte response markers, has the best prognosis amongst the subtypes, while CMS4, with increased expression of inflammatory and immunosuppressive markers, has the poorest. Despite their differences, CMS1 and CMS4 share some similarities in their response to chemotherapies [37]. CSF3 administration alongside chemotherapy has been shown to increase the risk of tumor regrowth in a model of breast cancer, although the same effect was not seen in a lung cancer mode [38]. CSF3 signaling may also play a role in chemotherapeutic response in CRC, either directly or through association with a particular molecular profile. In addition to associations with traditional chemotherapeutic agents, the CMS correlated with CSF3R may be the best candidates for immunotherapy, although their differing immune environments likely require different approaches [36]. CSF3R may be part of the therapeutic response signatures in the context of CMS1 or CMS4.

In addition to tumor cells, solid tumor tissue also contains immune, epithelial, and stromal cells. Signaling between these cells in the TME affects tumor progression, metastasis, and response to therapeutics. Transcriptome data from tumor tissue provides greater understanding of diverse tumor compositions and offers both prognostic value and vital insight into potential efficacy of treatment options. The ESTIMATE algorithm uses transcriptome data to infer stromal and immune cell fraction of tumor samples in the form of stromal, immune, and ESTIMATE scores [39]. The stromal and immune scores resulting from estimate analysis have signified survival rates and progression in certain cancer subtypes including GI cancers, with lower stromal, immune, and ESTIMATE scores indicating better patient prognosis [30]. Gene expression analysis of CRC patients revealed that isolated fibroblasts have the highest stromal scores, and CAFs have been shown to express high amounts of CSF3 [10, 30]. Therefore, it is not entirely surprising that we found that CSF3R is strongly associated with stromal scores. In the same study, leukocytes were found to have the highest immune scores. CSF3R is also strongly correlated with immune scores, suggesting a significant association between CSF3 signaling and changes in the immune milieu of the TME.

We found critical tumor-associated T cell populations and phenotypes were associated with CSF3 signaling. The positive relationship between Treg infiltration, and the negative correlation with naïve CD8 infiltration, are signs of a pro-tumor T cell environment. Tumor-specific cytotoxicity is challenging in CRC, as T cell infiltration is often both low and ineffective due to exhaustion [40, 41]. Treg populations, which produce immunosuppressive cytokines IL-10 and TGF-β, are associated with poor outcomes in CRC [41]. CD8 infiltration and activation are associated with positive outcomes in CRC [41, 42]. Naïve CD8 infiltrating T cells can become activated in the TME, proliferate, and contribute to anti-tumor responses [43]. These findings are consistent with our mouse model findings, where CSF3 inhibition decreased Treg and increased CD8 populations [14]. These data support a potential similar role in human CRC tumors. Interestingly, CSF3 signaling was associated with both negative and positive T cell proliferation signatures, but whether these pathways are co-activated in the same tumors

or activated in different subsets of tumors remains unknown. We also observed a shift in central and effector memory CD8 T cells associated with CSF3R in CC, but not RC, tumors. Others have found no connection between memory T cells and CSF3 treatment [44], suggesting that this association could be due to secondary factors in the TME shared between CSF3 signaling and memory formation.

In addition to associations with altered T cell signatures, CSF3R expression is associated with changes to innate immune cells in the TME. The role of tumor associated macrophages (TAM) in the setting of CRC remains a subject of intense study, given that there is evidence that macrophages can have pro-tumor or anti-tumor effects within the TME depending on phenotype [45–48]. As with T cells, we observed enrichment for both positive and negative regulation of macrophage proliferation, as well as finding an enrichment of both pro-inflammatory and anti-inflammatory macrophage signatures in CSF3R-high tumors. The mechanisms behind the multiple facets of macrophage behavior and tumor outcome are not completely understood, but tumor cell chemotactic factor excretion as well as regulation of the cytokine milieu are known to contribute [49, 50]. TAM can drive tumor proliferation and suppress T cell antitumor effects [40]. CRC macrophages do not align to classic conceptions of M1/M2 differentiation, and the full range of behaviors and their implications in cancer development and progression are still unknown [51]. Macrophages can be an abundant source of pro-inflammatory cytokines, including IL-6 and IL-1, which promote CRC [52, 53]. Additionally, macrophages help regulate CD8 cytotoxicity and can affect the efficacy of T cell-targeted immunotherapies. These aspects of macrophage behavior are potential therapeutic targets [54, 55]. CSF3 signaling can induce immunosuppressive behavior in macrophages, which could play a role in the altered T cell profiles reported here [56].

Our analysis of gene signatures suggested co-regulation of ERK/MAPK and WNT signaling pathways with CSF3 and CSFR. CSF3 is known to signal through ERK, and to prolong ERK activation [57, 58]. Interestingly, CSF3R was not correlated with ERK regulation signatures in CC, although it was correlated with MAPK cascade. The ERK/MAPK pathway promotes tumor cell proliferation and migration, and inhibition has shown some therapeutic potential in CRC cell lines with dysregulated pathways [59, 60]. CSF3 also correlated strongly with WNT5a expression, which can promote or inhibit tumor growth dependent on its isoform [61]. Although our findings do not shed light on which isoform is present in CRC, these findings are consistent with reports that WNT5a induces CSF3 [62]. Most CRCs develop slowly over years in a multistep process known as adenoma-carcinoma sequences [63–65]. Activating mutations of proteins involved in the Wnt signaling pathway are among the first in the CRC developmental process, the only known genetic variations present in early premalignant lesions, and thought to be responsible for the initiation of CRC [66]. Subsequent mutations in the proto-oncogene KRAS are often seen at the early adenoma stage of CRC development [67, 68]. The accompanying high Ras activity is associated with increased ERK/MAPK activity [69]. Correlations between CSF3/CSF3R signaling, Wnt signaling, and increased ERK/MAPK activity lead us to speculate that upregulation of CSF3/CSF3R signaling occurs in the early stages of CRC development. Further investigation into these pathways in the context of CC and RC is warranted.

The primary factors controlling CSF3 signaling in different subtypes of CRC are currently unknown. The percentage of CC and RC samples with mutations in either CSF3 or CSF3R was under 2%. Based on this, we believe the associations between higher CSF3 or CSF3R expression and other genes are related to amount of gene expression rather than mutations in CSF3 or CSF3R leading to upregulation of the associated genes. Additional regulation of the inflammatory milieu in tumors that express high CSF3/CSF3R may come from toll-like receptor (TLR). The correlation of TLR gene signatures with CSF3R also suggests potential enrichment

of TLR signaling. TLR signaling can amplify inflammation, which can drive tumor progression in CRC [70]. Their participation in CRC development and progression is not fully understood, but enrichment of some TLR pathways are associated with CRC progression [71]. Alongside the TLR signaling, both CC and RC tumors high in CSF3R were enriched in the IL-1β production pathway. IL-1 can regulate CSF3 production, but less is known about regulation of receptor expression [72].

One striking pattern that emerged from this review of the genes most strongly correlated with CSF3 and CSF3R was a marked association with invasion-associated proteins. Some of these we suspected would be present; for instance, CSF3 is associated with inflammation so we anticipated a correlation with CSF3 and IL-1 and IL-6. These cytokines are often upregulated in CRC and associated with many aspects of tumor progression and are under investigation as therapeutic targets [73–75]. However, we also found a cluster of chemokines and formyl peptide receptors that are implicated in invasion of tumors, as well as trafficking of immune cells. CXCL5, CXCL6, and CXCL8 bind to CXCR2, which induces ERK-regulated EMT [76, 77]. These chemokines have been shown to induce tumor metastasis in multiple solid tumors [78–80]. In addition, CXCL8 has been shown to be a predictor of poor response to some therapeutics in CRC [81]. Further investigation of these pathways in the context of CSF3 and CSF3R –upregulated tumors may provide insight into regulation of invasion.

CC and RC are often combined in the literature, but, due to anatomic location, are most commonly treated differently with all but the earliest stage rectal cancers generally subjected to chemoradiation, which may alter gene expression in the resected specimens, leading us to analyze them separately [18]. CC tumors are most commonly removed without prior treatment with therapeutic agents, so some of the differences observed when comparing results between CC and RC may be due entirely to pre-resection treatment. However, much of the data that we acquired showed similar results in both tumor types. This was true for established gene signatures, like the ESTIMATE scores, but we also observed many of the same genes significantly correlated with CSF3/R expression in both tumor types. While minimal shifts in Spearman ρ or p-values were observed, the direction and size effect were consistent with each other in many findings. Some apparent differences between CC and RC may be due to altered expression profiles (and possibly some observed similarities might not have been present before treatment of RC tumors). However, the large numbers of consistent results suggest that CSF3 signaling plays similar roles in both cancers. The same approach for both colon and rectal cancers could be used to direct future studies of CSF3 and CSF3R involvement in tumor development and progression.

This study has the standard limitations associated with assessment of the publicly available gene expression datasets, including the potential to have missed subtle differences based on race, environmental exposures, or specimen handling. Gene expression data also lacks some important characteristics of protein analysis, and some post-transcriptional modifications can affect cell behavior. The findings are also somewhat limited by the numbers of tumors in the database. We may have lost some biologically significant observations due to smaller effect size or higher variability between tumors. One potential trend that could be of interest is the proportion of men with high-CSF3 tumors is higher than the proportion of women with high-CSF3 tumors. Because CRC mortality is worse for men than women, we were intrigued to see a trend toward increased CSF3 expression in the tumors of men [82]. Although the findings did not reach significance, it is possible that a larger data set would provide insight. This is particularly intriguing given the receptor expression did not differ in tumors from men and women in CC, but there was a trend toward sex-related receptor expression in RC. Because of the relatively low number of RC tumors, this data is inconclusive. However, the association of

CSF3 and its receptor with many therapeutically relevant axes in CRC supports several hypotheses that can be tested with further mechanistic studies.

## Conclusion

CSF3 and CSF3R correlate with specific CMS subtypes (CMS1 and CMS4) and immune gene signatures that have survival implications in CRC. Furthermore, this cytokine and its receptor correlate with changes in immune cell types including CD8, Treg, and macrophage populations known to affect CRC outcomes. Analysis of gene expression reveals many enriched pathways in CSF3 or CSF3R high-expressing tumors, implicating CSF3 in immune cell trafficking and response, ERK signaling, and invasion in CC and RC tumors. The role that CSF3 and CSF3R play in the evolution of the CRC tumor microenvironment may offer a potential therapeutic target and should be further explored.

## Supporting information

**S1 Fig. CSF3R expression correlates with CSF3 expression in CC and RC.** Scatterplots depicting Spearman's correlation of TCGA Firehose Legacy dataset between CSF3R and CSF3 in CC (A) and RC (B).
(TIF)

**S2 Fig. CSF3 ligand and receptor expression is associated with IL-1 and IL-6.** Scatterplots depicting Spearman's correlation of TCGA Firehose Legacy dataset between IL-1α, IL-1β, or IL-6 and CSF3 in CC (A-C) and RC (D-F); and IL-1α, IL-1β, or IL-6 and CSF3R in CC (G-I) and RC (J-L). Red line indicates linear regression.
(TIF)

**S3 Fig. CSF3 ligand and receptor expression is associated with formyl peptide receptors (FPR).** Scatterplots depicting Spearman's correlation of TCGA Firehose Legacy dataset between FPR1 or FPR2 and CSF3 in CC (A, B) and RC (C, D); and FPR1 or FPR2 and CSF3R in CC (E, F) and RC (G, H). Red line indicates linear regression.
(TIF)

**S4 Fig. CSF3 ligand and receptor expression is associated with matrix metallopeptidases (MMPs).** Scatterplots depicting Spearman's correlation of TCGA Firehose Legacy dataset between MMP1, MMP3 or MMP10 and CSF3 in CC (A-C) and RC (D-F); and MMP1, MMP3 or MMP10 and CSF3R in CC (G-I) and RC (J-L). Red line indicates linear regression.
(TIF)

**S1 Dataset. Correlations between CSF3 and CSF3R in CC and RC.**
(XLSX)

## Acknowledgments

The results shown here are in whole or part based upon data generated by the TCGA Research Network: https://www.cancer.gov/tcga.

## Author Contributions

**Conceptualization:** Anita L. Ray, Katherine T. Morris.

**Data curation:** Apryl S. Saunders, Dawn E. Bender, Xiangyan Wu.

**Formal analysis:** Apryl S. Saunders, Dawn E. Bender, Xiangyan Wu.

**Funding acquisition:** Katherine T. Morris.

**Investigation:** Apryl S. Saunders, Dawn E. Bender, Xiangyan Wu.

**Methodology:** Apryl S. Saunders, Dawn E. Bender.

**Project administration:** Apryl S. Saunders, Anita L. Ray, Katherine T. Morris.

**Resources:** Xiangyan Wu, Katherine T. Morris.

**Supervision:** Apryl S. Saunders, Katherine T. Morris.

**Validation:** Apryl S. Saunders, Dawn E. Bender.

**Visualization:** Dawn E. Bender, Anita L. Ray, Xiangyan Wu.

**Writing – original draft:** Apryl S. Saunders.

**Writing – review & editing:** Apryl S. Saunders, Dawn E. Bender, Anita L. Ray, Katherine T. Morris.

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
