## [Decision Letter · Decision Letter 0]

22 Jan 2021

PONE-D-20-30842

Colony-stimulating factor 3 signaling in colon and rectal cancers: Immune response and CMS classification in TCGA data

PLOS ONE

Dear Dr. Morris,

Thank you for submitting your manuscript to PLOS ONE. After careful consideration, we feel that it has merit but does not fully meet PLOS ONE’s publication criteria as it currently stands. Therefore, we invite you to submit a revised version of the manuscript that addresses the points raised during the review process.

A number of issues have been raised in review which should be addressed if the authors plan to submit a revised manuscript.

We look forward to receiving your revised manuscript.

Kind regards,

Salvatore V Pizzo

Academic Editor

PLOS ONE

Journal Requirements:

Reviewers' comments:

Reviewer's Responses to Questions

**Comments to the Author**

1. Is the manuscript technically sound, and do the data support the conclusions?

Reviewer #1: Yes

2. Has the statistical analysis been performed appropriately and rigorously? 

Reviewer #1: Yes

3. Have the authors made all data underlying the findings in their manuscript fully available?

Reviewer #1: Yes

4. Is the manuscript presented in an intelligible fashion and written in standard English?

Reviewer #1: Yes

5. Review Comments to the Author

Reviewer #1: Thank you for the opportunity to review the manuscript "Colony-stimulating factor 3 signaling in colon and rectal cancers: Immune response and CMS classification in TCGA data." Please find my comments below:

1. TCGA is listed in the introduction line 72 for the first time without being defined. Please define for the reader. The same is true of ESTIMATE.

2. I find the use of noconventional abbreviations for colon and rectal cancer (COAD and READ) to be distracting. Why not CC and RC?

3. It seems as though you are implying that because colon and rectal cancers are treated somewhat differently they must be molecularly distinct. If this is indeed what you are implying it is an erroneous statement as rectal cancers are treated with radiation purely because of their location in the body. Likely colon cancer would have a similar response to irradiation, but this is not feasible due to it proximity to small bowel and the risk of radiation injury.

4. TCGA should be described in your materials and methods section.

5. Presumably dysregulation of the CF3 pathway will occur at different points in the mutational cascade for different tumors. Do you think that changes in the regulation of CSF3 in this cohort are the result of a primary mutation on the CSF3 and CSF3R genes or is this secondary to other genes/proteins effects on the regulation of CSF3 and CSF3R?

6.Are you able to determine at what point in tumorgenesis CSF3 becomes a key player in further tumor progression based on the ontological markers you have identified?

7. Your results section spends a significant amount of time explaining techniques like ESTIMTE and concepts like CMS1-4. These should be explained prior to the results section, either in the introduction or the methods and results should only be included in the results section.

6. PLOS authors have the option to publish the peer review history of their article (what does this mean?). If published, this will include your full peer review and any attached files.

Reviewer #1: No

---

## [Author Response · Author response to Decision Letter 0]

27 Jan 2021

Dear Dr. Pizzo,

We thank the editor and reviewer for their thoughtful review of our manuscript titled "Colony-stimulating factor 3 signaling in colon and rectal cancers: Immune response and CMS classification in TCGA data." We appreciate the insightful comments as well as the chance to resubmit the work. Below is an itemized response to the referee's comments. 

Journal Requirements

We have revised the manuscript to meet PLOS ONE’s style requirements, including those for file naming. 

Comment: TCGA is listed in the introduction line 72 for the first time without being defined. Please define for the reader. The same is true of ESTIMATE.

Response: We have defined these in the introduction (line 73) and expanded upon the definition of ESTIMATE in the methods section (line 76).

Comment: I find the use of nonconventional abbreviations for colon and rectal cancer (COAD and READ) to be distracting. Why not CC and RC?

Response: We have changed the abbreviations to CC and RC for colon and rectal cancer.

Comment: It seems as though you are implying that because colon and rectal cancers are treated somewhat differently they must be molecularly distinct. If this is indeed what you are implying it is an erroneous statement as rectal cancers are treated with radiation purely because of their location in the body. Likely colon cancer would have a similar response to irradiation, but this is not feasible due to its proximity to small bowel and the risk of radiation injury.

Response: We appreciate the reviewer’s comment and have changed the text in the Methods (line 93) and the Discussion (line 411) to reflect that we analyzed colon and rectal cancers separately due to concerns about the potential for significant changes in the transcriptomics due to preoperative chemoradiation therapy. 

Comment: TCGA should be described in your materials and methods section.

Response: We are grateful for the attention that was given to the materials and methods section. We have highlighted TCGA as a stand-alone material (line 87) to address this.

Comment: Presumably dysregulation of the CF3 pathway will occur at different points in the mutational cascade for different tumors. Do you think that changes in the regulation of CSF3 in this cohort are the result of a primary mutation on the CSF3 and CSF3R genes or is this secondary to other genes/proteins effects on the regulation of CSF3 and CSF3R?

Response: Thank you for drawing our attention to this excellent point. In TCGA data, the rate of mutations within the CSF3 or CSF3R genes in CRC samples is less than 2%. Based on that low rate of mutations in these genes, we believe observed associations are likely secondary to the influence of other tumor mutations, as well as the cytokine milieu, immune infiltrate, and stromal phenotype. We have published data from mouse models suggesting that in inflammation-associated cancers, CSF3 signaling is critical to tumor development at early stages, and increases during tumor development with no mutations in CSF3 or CSF3R. We have updated the Discussion (line 397) to include the frequency of mutations of CSF3 and CSF3R and reflect some of the other considerations.

Comment: Are you able to determine at what point in tumorigenesis CSF3 becomes a key player in further tumor progression based on the ontological markers you have identified?

Response: We thank the reviewer for this insightful comment. The correlations we observe between Wnt signaling and the ERK/MAPK activity with CSF3 and CSF3R leads us to believe that increase in CSF3 signaling occurs early in CRC development. We have added this to the Discussion (line 380).

Comment: Your results section spends a significant amount of time explaining techniques like ESTIMATE and concepts like CMS1-4. These should be explained prior to the results section, either in the introduction or the methods and results should only be included in the results section.

Response: We have adjusted the manuscript to include these descriptions in the introduction and methods section and appreciate the reviewer’s help in increasing the clarity of the work. 

We hope these alterations and responses have strengthened the manuscript and appreciate the chance to revise and resubmit this work.

Sincerely, 

Katherine Morris MD, FACS

---

## [Decision Letter · Decision Letter 1]

4 Feb 2021

Colony-stimulating factor 3 signaling in colon and rectal cancers: Immune response and CMS classification in TCGA data

PONE-D-20-30842R1

Dear Dr. Morris,

We’re pleased to inform you that your manuscript has been judged scientifically suitable for publication and will be formally accepted for publication once it meets all outstanding technical requirements.

Kind regards,

Salvatore V Pizzo

Academic Editor

PLOS ONE

Additional Editor Comments (optional):

Reviewers' comments:

Reviewer's Responses to Questions

**Comments to the Author**

1. If the authors have adequately addressed your comments raised in a previous round of review and you feel that this manuscript is now acceptable for publication, you may indicate that here to bypass the “Comments to the Author” section, enter your conflict of interest statement in the “Confidential to Editor” section, and submit your "Accept" recommendation.

Reviewer #1: All comments have been addressed

2. Is the manuscript technically sound, and do the data support the conclusions?

Reviewer #1: Yes

3. Has the statistical analysis been performed appropriately and rigorously? 

Reviewer #1: Yes

4. Have the authors made all data underlying the findings in their manuscript fully available?

Reviewer #1: Yes

5. Is the manuscript presented in an intelligible fashion and written in standard English?

Reviewer #1: Yes

6. Review Comments to the Author

Reviewer #1: (No Response)

7. PLOS authors have the option to publish the peer review history of their article (what does this mean?). If published, this will include your full peer review and any attached files.

Reviewer #1: No

---

## [Editor Report · Acceptance letter]

11 Feb 2021

PONE-D-20-30842R1 

Colony-stimulating factor 3 signaling in colon and rectal cancers: Immune response and CMS classification in TCGA data 

Dear Dr. Morris:

I'm pleased to inform you that your manuscript has been deemed suitable for publication in PLOS ONE. Congratulations! Your manuscript is now with our production department. 

Kind regards, 

on behalf of

Dr. Salvatore V Pizzo 

Academic Editor

PLOS ONE